# The Complicated Relationship of Short-Chain Fatty Acids and Oral Microbiome: A Narrative Review

**DOI:** 10.3390/biomedicines11102749

**Published:** 2023-10-11

**Authors:** Georgy E. Leonov, Yurgita R. Varaeva, Elena N. Livantsova, Antonina V. Starodubova

**Affiliations:** 1Federal Research Center of Nutrition, Biotechnology and Food Safety, 109240 Moscow, Russia; varaeva@ion.ru (Y.R.V.); medeliux@gmail.com (E.N.L.); avs.ion@yandex.ru (A.V.S.); 2Therapy Faculty, Pirogov Russian National Research Medical University, 117997 Moscow, Russia

**Keywords:** short-chain fatty acids, oral microbiota, gut microbiota, oral–gut microbiome axis, biofilms, oral diseases, systemic diseases, periodontitis, oral cancer, chronic inflammation

## Abstract

The human oral microbiome has emerged as a focal point of research due to its profound implications for human health. The involvement of short-chain fatty acids in oral microbiome composition, oral health, and chronic inflammation is gaining increasing attention. In this narrative review, the results of early in vitro, in vivo, and pilot clinical studies and research projects are presented in order to define the boundaries of this new complicated issue. According to the results, the current research data are disputable and ambiguous. When investigating the role of SCFAs in human health and disease, it is crucial to distinguish between their local GI effects and the systemic influences. Locally, SCFAs are a part of normal oral microbiota metabolism, but the increased formation of SCFAs usually attribute to dysbiosis; excess SCFAs participate in the development of local oral diseases and in oral biota gut colonization and dysbiosis. On the other hand, a number of studies have established the positive impact of SCFAs on human health as a whole, including the reduction of chronic systemic inflammation, improvement of metabolic processes, and decrease of some types of cancer incidence. Thus, a complex and sophisticated approach with consideration of origin and localization for SCFA function assessment is demanded. Therefore, more research, especially clinical research, is needed to investigate the complicated relationship of SCFAs with health and disease and their potential role in prevention and treatment.

## 1. Introduction

Nowadays, the human body microbiome is causing an increasing research interest [1]. The number of original papers devoted to the investigation of biota-associated axes has rocketed in the past decades due to the development of new ‘omics’ technologies. At the same time, the high prevalence of oral cavity pathology, including dental caries, periodontitis, and oral cancer, is also causing great concern worldwide [2]. Thus, the new focus for research is the oral flora, which has physiological, metabolic, immunological, mucosal protective, nutritional, and detoxification functions [3].

The human oral microbiome is composed of bacteria, archaea, viruses, fungi, and protozoa [4]. The oral cavity harbors 774 bacterial species, according to the expanded Human Oral Microbiome Database (HOMD), with 58% having formal names, 16% being unnamed but cultivated, and 26% known solely as uncultivated phylotypes [5]. Predominantly, *Actinobacteria* (genera *Corynebacterium*, *Rothia*, and *Actinomyces*), *Bacteroidetes* (genera *Prevotella*, *Capnocytophaga*, and *Porphyromonas*), *Firmicutes* (genera *Streptococcus* and *Granulicatella*), *Fusobacteria* (genus *Fusobacterium*), and *Proteobacteria* (genera *Neisseria* and *Haemophilus*) are found in the oral microbiomes of healthy individuals [6]. Furthermore, there is a reliable relationship between oral microbiota and systemic diseases, including inflammatory bowel disease (IBD), cancers, cardiovascular diseases, Alzheimer’s disease, diabetes, rheumatoid arthritis, and preterm birth [7,8,9,10,11,12,13].

The species diversity of oral microorganisms is influenced by various factors such as host genetics, gender, age, oral health, smoking, diet, socioeconomic factors, and medications [14,15,16,17,18,19,20].

Chronic inflammation and oxidative stress play a critical role in the pathogenesis of damage in both systemic and localized oral diseases [21]. Short-chain fatty acids (SCFAs) have an important anti-inflammatory function in the regulation of immune activity, contributing to the prevention of several chronic inflammatory conditions [22]. They can directly influence neutrophils by decreasing their production of reactive oxygen species (ROS) and myeloperoxidase (MPO) while also enhancing their apoptosis [23,24]. SCFAs, also known as volatile fatty acids, consist of organic linear carboxylic acids containing fewer than six carbons, which include formic acid (C1), acetic acid (C2), propionic acid (C3), isobutyric and butyric acids (C4), and isovaleric and valeric acids (C5) [25]. These SCFAs are generated by the bacterial fermentation of dietary fiber and resistant starch. Butyrate, propionate, and acetate make up approximately 90% to 95% of the SCFAs found in the colon [26].

The composition of a healthy gut microbiota is primarily composed of two phyla, *Firmicutes* and *Bacteroidetes*, accounting for nearly 90%, alongside less-represented phyla like *Proteobacteria*, *Verrucomicrobia*, or *Actinobacteria*. Acetate and propionate are the primary products of the *Bacteroidetes* phylum, while *Firmicutes* produces butyrate as its primary metabolic end product. SCFAs contribute to the integrity and permeability of the gut barrier through various mechanisms [27]. Primarily, butyrate increases the concentration of tight junction proteins, such as claudin-1 (CLDN1), zonula occludens-1 (ZO-1), and occludin, by activating the gene expression for encoding these proteins [28]. Additionally, butyrate reinforces the mucosal layer of the intestinal epithelium by upregulating the expression of mucin 2 [29].

SCFAs play a crucial role in regulating various physiological pathways within the nervous system. They modulate systemic and neuroinflammation by influencing the functions and structures of microglial cells, which subsequently impact emotions, cognitive functions, and mental disorders. Furthermore, it appears that high concentrations of SCFAs are associated with the core expression of neurotrophic factors. [30,31]. SCFAs, especially acetate, also play a critical role in appetite and human metabolism regulation [32]. Animal models have shown that diets high in fermentable carbohydrates, which are broken down in the colon to produce SCFAs, lead to appetite reduction [33]. SCFAs are the host’s energy metabolites that provide nearly 10% of the daily energy requirements for humans [34].

The mentioned data are not only theoretically valuable but also applicable to resolving various human health issues. The involvement of short-chain fatty acids in oral microbiome composition, oral health, and chronic inflammation, as well as the potential development of SCFA-based biomarkers and therapies aimed at improving oral microbiome health, oral disease, and chronic inflammation, are gaining increasing attention. To analyze the problem, we conducted a search on PubMed/MEDLINE, SCOPUS, and Google Scholar with the following MESH terms/keywords: “Oral microbiota AND short-chain fatty acids”, “Oral microbiota AND (oral diseases OR periodontitis OR oral cancer OR systemic diseases)”. In this narrative review, identified results of early in vitro, in vivo, and pilot clinical studies and research projects are presented in order to define the boundaries of this new complicated issue.

## 2. SCFAs and Oral Health

The oral bacteria are not pure planktonic in saliva but consist primarily of a structured community of aggregated bacterial cells (either from the same species or multi-species) embedded and enclosed in a self-produced extracellular polymeric matrix and adherent to an inert or living surface (the biofilms) [35]. Biofilms occur in the oral cavity as a normal physiological process, as well as in the colon or female reproductive tract, notwithstanding that biofilms are commonly associated with human pathology [36]. Inside biofilms, there are gradients of nutrients, signaling compounds, and bacterial waste products that result from environmental conditions and the physiological responses of bacteria to the local environment. Thus, within a consortium, bacteria can achieve multicellular metabolic coordination and expansion by utilizing additional metabolic pathways or engaging in cross-feeding of metabolic intermediates [37,38]. These adaptive responses heavily rely on the perception and processing of chemical information from the environment, which plays a central role in their regulatory control. Microorganisms that establish colonies in the oral cavity possess a diverse range of adaptive mechanisms encoded in their genes, enabling the oral microbiota to seamlessly adapt and integrate into an oral biofilm lifestyle. This adaptation involves optimizing their phenotypic characteristics to suit the specific environment [39]. As a consequence, microorganisms within biofilms exhibit distinct phenotypic traits compared to their planktonic counterparts.

The segregation of the oral microbiome into different niches can be attributed to several factors, including pH, salinity, redox potential, oxygen levels, and nutritional availability [40]. The microbial composition varies across various regions of the oral cavity, including the gingival sulcus, tongue, cheek, hard and soft palate, floor of the mouth, throat, saliva, and teeth, with each area presenting unique characteristics [41]. These interconnected intra-oral habitats can harbor different microbial profiles, collectively forming the oral microbiome as a meta-microbial community. Moreover, based on cluster analysis of microbial genera distribution in different oral niches, three distinct ‘metaniches’ with similar compositions have been identified. These metaniches are categorized based on the composition of bacteria and include the following: supragingival plaque and gingival crevicular fluid (P-GCF); saliva-tongue-hard palate (S-T-HP); and cheek and sublingual area (C-U). In saliva, the predominant species include *Streptococcus*, *Prevotella*, *Veillonella*, *Neisseria*, and *Haemophilus*. At the genus level, *Streptococcus*, *Gemella*, *Veillonella*, *Haemophilus*, *Neisseria*, *Porphyromonas*, *Fusobacterium*, *Actinomyces*, and *Prevotella* are consistently found across all healthy human oral sites, regardless of their niche location. Notably, *Streptococcus* is the dominant genus, followed by *Haemophilus* in the buccal mucosa, *Actinomyces* in the supragingival plaque, and *Prevotella* in the subgingival plaque. In the “P-GCF” metaniche, strong positive correlations were observed between cytokine levels and the relative abundance of genera such as *Aggregatibacter*, *Fusobacterium*, *Gemella,* and *Streptococcus*, while negative correlations were evident with *Tannerella, Leptotrichia*, *Corynebacterium*, *Capnocytophaga*, *Saccharimononadae,* and *Neisseriacae*. In the “S-T-HP” metaniche, *Gemella spp*. exhibited the highest positive correlations with the cytokine levels, and variable positive connections were identified for the genera *Capnocytophaga*, *Streptococcus*, *Porphyromonas*, and *Campylobacter* [42].

Within the oral microbiota, there exists a substantial number, approximately 1839, of biosynthetic gene clusters responsible for generating a diverse array of metabolites through various mechanisms [43]. SCFAs can be synthesized in the oral cavity using two primary approaches: carbohydrate hydrolysis or amino acid metabolism. Carbohydrates undergo fermentation, leading to the formation of monosaccharides, followed by the conversion to pyruvate through either the pentose phosphate pathway (C5) or glycolysis (C6) [44]. Bacterial species with the capability to convert sugars into SCFAs include *Streptococcus*, *Actinomyces*, *Lactobacillus*, *Propionibacterium*, and *Prevotella* [45]. In addition to food sources, oral bacteria can use salivary mucins, which cover the surface of the oral mucosa and contain glycans, as a nutrient source after they are fermented by sialidases [46]. Amino acids also offer a substrate for SCFA production through the process of transamination. The microbiota employs proteases and peptidases to break down proteins, and subsequent deamination of the resulting peptides and amino acids yields SCFAs in the oral cavity. *Actinomyces*, *Veillonella*, and *Fusobacterium* predominantly serve as the primary proteolytic bacteria [47]. Typical concentrations of SCFAs in the human oral cavity fall within the ranges of 6.3–16.2 mM for acetate, 1.2–3.1 mM for propionate, and 0.0–0.4 mM for butyrate [48]. The nature of the relationship between oral commensals, whether mutualistic or competitive, can be determined by the antimicrobial potential of these SCFAs. The antimicrobial potential of SCFAs can alter the structure of the oral microbiome to both support and inhibit bacterial growth [49].

Thus, high dietary carbohydrate intake leads to the proliferation of saccharolytic and acidogenic bacteria in supragingival plaques due to different strains. Eventually, this may cause the formation of ecosystems with increased cariogenic potential, which contributes to the demineralization of tooth surfaces and the development of oral diseases [50]. A recent study showed that the concentrations of SCFAs, as well as acetate alone, correlate with the abundance of the following bacterial species: *T. denticola*, *Treponema socranskii*, *Filifactor alocis*, *T. forsythia*, *P. gingivalis*, *Porphyromonas endodontalis*, *Prevotella dentalis*, and *F. nucleatum*. Butyrate has been linked to *T. denticola*, *F. alocis*, *T. socraskii*, *F. nucleatum*, *T. forsythia*, and *P. gingivalis*, while propionate has only been associated with *T. denticola* and *F. alocis* [51]. The data on certain oral bacteria production of SCFAs is given in Table 1.

Elevated concentrations of SCFAs have been associated with soft tissue damage and an increased inflammatory response within the oral cavity [55]. Substantial and reliable evidence suggests that SCFAs can exert adverse effects on periodontal cells [56]. Notably, exposure to SCFAs has been observed to induce oral epithelial cell death, primarily through pyroptosis and apoptosis mechanisms [57]. In an in vitro model, it was observed that acetate, propionate, and butyrate, each at a concentration of 12.5 mM, induced cell death. Propionate exhibited the highest level of toxicity, while acetate displayed the lowest toxicity levels [58,59]. Moreover, millimolar concentrations of butyrate were found to impair the integrity of the gingival epithelial barrier, resulting in the disruption of cell–cell junctions, an upregulation of caspase-3 and gasedermin-E expression, and the induction of cell pyroptosis [60]. Furthermore, the study revealed that treatment with butyrate and propionate led to an increased relative abundance of keratin proteins in gingival epithelial cells, particularly keratin K17 [61]. Additionally, butyrate demonstrated the capacity to reduce the cytokine-induced expression of adhesion molecule 1 (ICAM-1) in epithelial cells [62]. In addition, SCFAs have been found to suppress the expression of various components, including connexin 26 and 43, cadherin-1, adhesion molecule-1, claudin-1 and 4, as well as desmoglein-1 and desmocollin-2. Such suppression can result in increased epithelial permeability [63].

Oxidative stress has also been implicated in epithelial cell damage, particularly in gingival fibroblasts. Prolonged butyrate exposure to gingival fibroblasts may lead to cell apoptosis, an increase in BAX expression, and the activation of caspase 8 and caspase 9 [64,65]. Moreover, an in vivo study demonstrated that prolonged treatment with butyrate led to iron non-accumulation, ROS formation, glutathione depletion, lipid peroxidation, and eventual ferroptosis in periodontal pocket tissues. It should be highlighted that butyrate also plays a role in stimulating T-cell apoptosis, promoting osteoblast differentiation, and releasing pro-inflammatory cytokines from neutrophils [66].

Nevertheless, several studies have indicated the situational benefit of SCFAs. For instance, the reduction in local immune response caused by long-term antibiotic treatment of *Candida albicans* infections was partially compensated by the application of SCFAs. These fatty acids have been suggested to inhibit *C. albicans* growth and stimulate the expression of Foxp3 and IL-17A in CD4+ T cells [67]. Therefore, SCFAs play a complicated role in oral cavity health. SCFAs are synthesized by the oral microbiota as a part of normal local metabolic processes, whereas the increased formation of SCFAs may be considered a marker of oral dysbiosis. Furthermore, excess SCFAs participate in local oral cavity pathology development as well as in pro-inflammatory signaling.

## 3. SCFAs and Oral Diseases

Increased levels of SCFAs have been linked to the development of numerous oral diseases, including dental caries, periodontitis, oral cancer, and various viral infections [68]. SCFAs take part in the regulation of bacterial growth by reshaping the microbiome structure, thereby promoting the proliferation of predominantly periodontal pathogens. Furthermore, specific SCFAs contribute to the formation of biofilms on the oral cavity’s surface [69]. The mentioned harm inflicted on oral tissues, primarily by butyrate and propionate, also contributes to the progression of oral diseases. A common underlying mechanism associated with oral lesions involves an intensified inflammatory response and heightened oxidative stress [70]. Acetate, propionate, and butyrate have been identified as ligands for transmembrane G-protein receptors known as FFAR2 (GPR43), FFAR3 (GPR41), and GPR109a [71]. These receptor types are expressed in cells throughout the body and serve various functions. Among the diverse SCFAs, FFAR2 shows a preference for acetate and propionate as ligands, while FFAR3 primarily binds to butyrate. For instance, one of the functions of FFAR2 is to signal about excess energy intake, inhibit insulin signaling in adipocytes, and thus reduce fat accumulation in adipose tissue [72]. Also, activation of FFAR3 induces changes in the hematopoietic activity of the bone marrow, characterized by an increased production of macrophages and dendritic cell precursors. As a result, highly phagocytic dendritic cells can infiltrate the lung tissue during allergic inflammation, for instance, where they establish the immune microenvironment and influence the severity of the reaction [73]. Notably, within the context of the oral cavity, the FFAR2 receptor is expressed in various immune cell types, including monocytes, neutrophils, eosinophils, and regulatory T cells (Treg). The finding that FFAR2 activation, particularly through SCFA administration such as acetate, confers resistance against several bacterial and viral infections is of particular significance [74]. A common feature observed in these SCFA-mediated processes is the direct or indirect stimulation of leukocytes via FFAR2 activation [75]. The detailed data of some recent studies investigating the association of SCFAs with oral health and disease is presented in Table 2.

Describing the contribution of SCFAs to the pathogenesis of oral cavity diseases, it should be considered that they are generally multifactorial pathologies, and SCFAs participate in the development of the diseases as an element of the sophisticated pathological chain consistent with the list of numerous factors working in joint action.

### 3.1. SCFAs and Dental Caries

Dental caries, a chronic infectious disease, results in the demineralization of dental hard tissues. This demineralization arises from a complex interplay between the resident flora and fermentable carbohydrates found in plaque [76]. The pathological processes associated with caries involve numerous microorganisms, including *Actinomyces gerencseriae*, *Bifidobacterium*, *S. mutans*, *Veillonella*, *S. salivarius*, *S. constellatus*, *S. parasanguinis*, and *Lactobacillus fermentum* [77]. 

**Table 2 biomedicines-11-02749-t002:** Recent studies investigating the association of SCFAs with oral health and disease.

Disease	Authors	Year	SCFAs	Population	Brief Results
Periodontitis	K Hatanaka et al. [78]	2022	C2 and C3–C6	10 healthy participants and 10 participants with mild and severe periodontal disease	No significant difference in saliva was observed between healthy participants and patients. However, C3–C6 significantly differed between mild and severe periodontal disease.
	JC Provenzano et al. [79]	2014	All	18 adult patients (ages ranging from 20 to 39 years) with asymptomatic apical periodontitis	Both propionate and butyrate were found in most of the root canal tissue samples examined. Before treatment, the predominant bacteria were *F. nucleatum* and members of the *Actinobacteria phylum*, and after treatment, *Streptococcus*.
	OJ Park et al. [80]	2023	Butyrate	THP-1 cell line and six-week-old female Sprague Dawley rats	Butyrate combined with lipoteichoic acid significantly increased caspase-1 activation and IL-1β secretion both in vitro and in vivo in a rat model of apical periodontitis.
	AD Rudin et al. [81]	2021	All	Primary human neutrophils	*P. gingivalis* from both lab strains and clinical specimens has been shown to produce significant amounts of SCFAs. A similar mixture of SCFAs induces Ca^2+^ signaling and chemotaxis in human neutrophils through activation of FFAR2.
	L. Qiqiang et al. [82]	2012	All	37 individuals (21 patients with chronic periodontitis and 16 periodontally healthy controls)	Periodontal treatment decreased the concentration of lactic, propionic, butyric, and isovaleric acids in the gingival crevicular fluid to the level of a healthy control group. The formic acid concentration increased. A rebound effect was observed for all SCFAs within 2–6 months.
	R Lu et al. [48]	2014	All	40 individuals (20 with generalized aggressive periodontitis, 20 healthy controls)	Formic acid concentration increased significantly, and acetic, propionic, and butyric acid concentrations decreased after conservative treatment of periodontitis. Sites containing *P gingivalis*, *T denticola*, *P intermedia,* or *F. nucleatum* showed a similar correlation.
	N Murakami et al. [83]	2022	Butyrate	Male C57BL/6 mice	Butyrate modulates periodontal mechanical nociception via FFAR3 signaling in P. gingivalis-induced periodontitis.
	ME Cueno et al. [84]	2018	Butyrate	10-week-old male Wistar rats	Administration of butyrate (5 mM) resulted in increased NADPH-related oxidative stress and inflammation, presumably mediated by MMP-9, in a rat model of periodontitis.
	S Ji et al. [85]	2023	All	16 individuals (8 with periodontitis, 8 healthy controls)	A total of 570 human proteins associated with inflammation, cell death, and metabolism were found to be differentially expressed in periodontitis. Microbial proteins associated with butyrate metabolism were upregulated in the periodontitis group.
	R Lu et al. [86]	2013	All	34 individuals (20 with generalized aggressive periodontitis, 14 healthy controls)	Patients with periodontitis had significantly higher concentrations of succinic, acetic, propionic, butyric, and isovaleric acids and a higher -abundance of both *P. gingivalis* and *T. denticola*. The level of SCFAs correlated positively with the number of these bacteria.
	HS Na et al. [51]	2021	All	112 individuals (79 with periodontitis, 33 healthy controls)	SCFAs correlated with the abundance of the following bacterial species in the periodontitis cohort: *T. denticola*, *Treponema socranskii*, *Filifactor alocis*, *T. forsythia*, *P. gingivalis*, *Porphyromonas endodontalis*, *Prevotella dentalis*, and *F. nucleatum.*
	AD Rudin et al. [87]	2021	Acetate and Butyrate	Primary human neutrophils	Predominantly acetate and butyrate, which are released in large amounts as end products of *F. nucleatum* metabolism, induce human neutrophil chemotaxis and cytosolic Ca 2+ signals via the FFAR2 receptor.
Oral cancer	Z Nouri et al. [88]	2023	All	309 adult cancer patients and 745 healthy controls	*Leuconostoc*, *Streptococcus*, *Abiotrophia,* and *Prevotella* were decreased in the cancer group, while *Haemophilus* and *Neisseria* were increased. Total SCFA and FFAR2 expression levels were higher in the control group, while TNFAIP8, IL6, and STAT3 levels were higher in the cancer group.
	Y Miyazaki et al. [89]	2010	Butyrate	Ca9-22, HSC-2, -3, and -4 cells	Butyrate induced the expression of podoplanin in HSC-2 and -3 cells and vimentin in Ca9-22 cells. Cell migration was stimulated at low concentrations of butyrate, especially in HSC-3 cells, while it was inhibited in HSC-2 and -4 cells.
HIV	K Imai et al. [90]	2009	Butyrate	ACH-2 and U1 cells	*P. gingivalis* produces high levels of butyric acid, which acts as an HDAC inhibitor and induces histone acetylation, leading to the induction of RNA polymerase II and the reactivation of HIV-1.
Epstein–Barr virus	K Imai et al. [91]	2012	Butyrate	Daudi cell line	The butyric acid of P. gingivalis stimulates the expression of ZEBRA and inhibits HDAC, which results in increased histone acetylation, increased activity of the BZLF1 gene transcription, and the induction of EBV reactivation.
KSHV	X Yu et al. [92]	2014	All	BCBL1 cells	SCFAs dose-dependently and synergistically stimulate lytic expression of KSHV genes. By transactivating viral chromatin, SCFAs inhibit HDAC, suppress the expression of SIRT1, EZH2, and SUV39H1, increase acetylation, and decrease histone trimethylation.
Dental caries	J Wu et al. [93]	2022	Acetate	Biofilm in vitro	*Lactobacillus casei* in a multi-species biofilm has a competitive advantage over *S. mutans* due to acetate production.
	T Park et al. [94]	2021	All	Biofilm in vitro	A mix of SCFAs inhibits *S. gordonii* biofilm formation by downregulating comD and comE mRNAs, which are regulators of the CSP pathway.

The extent of contribution by oral bacteria to dental caries hinges on metabolic pathways that encompass the absorption and metabolism of carbohydrates, including the production of SCFAs. Among these microorganisms, *S. mutans* plays a pivotal role as a major initiator of dental caries. This is due to its capacity to synthesize exopolysaccharides, metabolize carbohydrates into organic acids, particularly lactic acid, and maintain a low pH environment within the oral cavity [93]. SCFAs, as organic acids, also contribute to the environment conducive to the proliferation of acidogenic bacteria. On the other hand, the most relevant producers of SCFAs in the oral cavity are generally periodontal pathogens such as *P. gingivalis* and *F. nucleatum*. These bacteria compete with the main saccharolytic bacteria and actually increase the pH of the environment through amino acid metabolism, whereas they also directly inhibit the growth of cariogenic bacteria caused by SCFAs [95].

Studies centered on oral biofilms have revealed that acetate, propionate, and butyrate exhibit effective inhibitory activity against the formation of *S. gordonii* biofilms [94]. Furthermore, research has demonstrated that *Lactobacillus casei* possesses the capability to suppress the growth of *S. mutans*, including the conversion of lactate into acetate [95]. While it has been hypothesized that *Veillonella’s* conversion of lactic acid, produced by streptococci, into less potent acids like acetic acid may reduce the host’s susceptibility to caries, experimental evidence supporting this notion is lacking. Conversely, a molecular study suggests that *Veillonella* coexists with *Streptococcus* in carious lesions [96]. Understanding the complex relationships between oral bacteria and their metabolic pathways is essential for developing effective strategies for dental caries prevention and management.

### 3.2. SCFAs and Periodontitis

Periodontitis is a chronic inflammatory disease characterized by the destruction of the alveolar bone and periodontal tissues, leading to clinical manifestations such as deepening gum pockets and tooth loss. Early signs of periodontitis may involve bleeding during tooth-brushing, tooth mobility during eating, and conditions like halitosis (bad mouth odor), with additional signs emerging at later stages [97]. The dysbiosis of the oral microbiome, particularly within the periodontal pocket, represents a primary factor in the pathogenesis of periodontitis [98]. The most important periodontal pathogens are Gram-negative rod-shaped bacteria such as *P. gingivalis*, *P. intermedia*, *T. forsithia*, *T. denticola,* and *F. nucleatum* [99]. The development of periodontal diseases is significantly influenced by the formation of biofilms. Butyrate has been shown to stimulate fimbrilin-dependent colonization of *Actinomyces oris* and promote biofilm growth in the early stages of biofilm formation [100]. Higher expression of enzymes involved in *F. nucleatum* butyrate production is observed during biofilm formation compared to planktonic growth, suggesting the importance of butyrate as a virulence factor for certain bacteria [101]. SCFA-producing bacteria such as *Porphyromonas gingivalis*, *Tannerella forsythia*, *Prevotella intermedia,* and *Treponema denticola* also contribute to the multi-species expansion of biofilms [102]. Research has demonstrated significant alterations in local concentrations of SCFAs among subjects with periodontitis. Specifically, in the gingival crevicular fluid of individuals with chronic periodontitis, notably higher levels of propionate (11.68 ± 8.84 vs. 5.87 ± 3.35 mM) and butyrate (3.11 ± 1.86 vs. 1.10 ± 0.87 mM) have been observed when compared to their healthy counterparts. Furthermore, individuals with aggressive periodontitis exhibited elevated concentrations of acetic, propionic, and butyric acids (26.0 versus 11.3, 8.8 versus 2.1, and 2.5 versus 0.0 mM, respectively) [48]. The evidence suggests that periodontal treatment leads to a reduction in the concentrations of lactic, propionic, butyric, and isovaleric acids in the sulcular fluid, effectively returning them to levels comparable to those of a healthy control group. Although the post-treatment concentrations of formic acid tend to increase, these effects persist for several months [82]. Patients with periodontitis have significantly higher concentrations of succinic, acetic, propionic, butyric, and isovaleric acids, as well as a predominance of both *P. gingivalis* and *T. denticola*, compared to healthy individuals. The level of SCFAs positively correlates with the number of these bacteria [86]. In addition to the periodontal pocket, SCFAs, including propionate and butyrate, were detected in tissue samples from the root canals during the treatment of apical periodontitis. It should be pointed out that prior to treatment, the predominant bacterial species included *F. nucleatum* and members of the *Actinobacteria phylum*, but after treatment, representatives of the *Streptococcus* genus were prevalent [79]. In a study on a murine periodontitis model, butyrate was shown to modulate periodontal mechanical nociception through FFAR3 signaling, which may explain the often observed absence of pain in periodontitis [83]. Studies in vitro have shown that mainly acetate and butyrate, which are released in large amounts as metabolic end products of *F. nucleatum* and *P. gingivalis*, induce human neutrophil chemotaxis and cytosolic Ca^2+^ signals via the FFAR2 receptor [81,87]. However, SCFAs in the oral cavity may not only act locally; a recent study in rats found that butyrate injected directly into the gum was able to enter the systemic circulation and adversely affect distant organs [84]. To summarize, SCFAs play a critical role in periodontitis pathogenesis.

### 3.3. SCFAs and Oral Cancer

Oral cancer comprises a substantial portion of head and neck cancers, accounting for 48%, with approximately 90% of oral cancer cases being histologically classified as oral squamous cell carcinoma [103]. Acidogenic and aciduric species play a significant role in facilitating the invasion and metastasis of malignant cells. This is achieved through the promotion of an acidic tumor microenvironment via acid production [104]. Numerous in vitro studies have underscored the significance of acidic conditions in amplifying the metastatic potential of tumor cells [105]. Furthermore, SCFAs have been implicated in inhibiting the development of an effective immune response against tumor cells. They accomplish this process by attracting myeloid suppressive cells, thus expanding the population of immunosuppressive cells within the tumor microenvironment [106]. Butyrate has been observed to suppress the cytokine-induced expression of intercellular ICAM-1 on oral squamous carcinoma cells [62], which results in the promotion of leukocyte transmigration to pathologically affected areas and the activation of macrophage polarization toward the M2 tumor phenotype [107].

Human studies indicate a potential connection between the oral microbiota and carcinogenesis in several organs, including the gastrointestinal tract, head and neck, oral cavity, and pancreas. Moreover, recent research on the genera *Lactobacillus* and *Streptococcus* has identified the production of volatile sulfur compounds, SCFAs, reactive oxygen species, reactive nitrogen species, hydrogen peroxide, and lactate, which are associated with carcinogenesis, chronic inflammation, genomic instability, and tumor angiogenesis [88]. Also, investigations have shown that *P. gingivalis* promotes the development and progression of oral cancer by promoting oral cell proliferation and inducing the expression of key molecules such as nuclear factor kappa B (NF-κB), IL-6, signal transducer and activator of transcription 3 (STAT3), cyclin D1, and matrix metallopeptidase 9 (MMP-9) [108]. The relationship between *F. nucleatum* and the development of oral cancer has been extensively researched. Epidemiological studies indicate that *F. nucleatum* is the most prevalent species colonizing oral mucosal sites affected by tumors. Although *F. nucleatum* is positively correlated with oral cancer, the bacterium is specifically less common in advanced cancers and is also associated with better survival [109]. *F. nucleatum* is a significant producer of SCFAs, especially propionate and butyrate, and while there has been previous evidence of *F. nucleatum*’s ability to activate neutrophils via the FFAR2 pathway, there is no direct evidence of *F. nucleatum’s* effect on cancer development via SCFAs [110]. In a recent study investigating the correlation between oral microbiota, their metabolites, and the development of different cancer types, it was discovered that cancer patients had lower levels of *Leuconostoc*, *Streptococcus*, *Abiotropia*, and *Prevotella*. Conversely, the concentration of *Haemophilus* and *Neisseria* increased in this group. The control group presented higher expression levels of SCFA and FFAR2, while the cancer group had higher levels of TNFAIP8, IL6, and STAT3 [88]. Therefore, early-stage studies suggest that some SCFA-producing bacteria may promote cancer development in the oral cavity. However, SCFAs’ effects are associated with possible activation of immune suppression of carcinogenesis.

### 3.4. SCFAs and Viral Disease

The oral cavity is a site of active viral replication and shedding, including human immunodeficiency viruses (HIV) and herpesviruses, Kaposi’s sarcoma-associated herpesvirus (KSHV), and Epstein–Barr virus (EBV) [111]. Although these viruses are the causative agents of their respective infections, the oral microbiome can have a significant impact on them. Disruption of host-microbial homeostasis in the epithelial tissues of the oral cavity contributes to viral replication [112], which is characterized by a change in the composition of the polymicrobial community of the oral cavity towards a dysbiotic and often pathogenic one. Consequently, the described shifting contributes to overactivation of the immune system and inflammatory conditions. *S. mutans*, *Lactobacillus*, *Candida*, *Haemophilus parahaemolyticus*, *Actinomyces*, *Neisseria subflava*, and *Corynebacterium diphtheriae* species are more prevalent in the saliva of HIV patients, while Streptococcus mitis is less frequently found [113]. Furthermore, the potential increase in opportunistic bacterial load, which can facilitate HIV reactivation and expedite the progression of AIDS, exists in the context of AIDS-associated immunosuppression. Notably, periodontitis-associated bacteria-produced SCFAs have demonstrated the capability to reactivate KSHV. This reactivation process is mediated through the inhibition of histone deacetylase (HDAC) and the downregulation of various epigenetic modulators (EZH2, SUV39H1) responsible for histone trimethylation repression. As a result, exposition of SCFAs to KSHV-infected cells exhibits histone hyperacetylation, thereby facilitating the transactivation of viral chromatin [92]. Consequently, individuals afflicted with periodontal diseases may exhibit increased lytic reactivation of KSHV and a higher risk of the development of oral Kaposi’s sarcoma. Comparable effects of butyrate were described for Epstein–Barr virus activation [91].

Overall, in some medical conditions, viruses can replicate in the oral cavity, inducing complicated relationships with the oral microbiome. SCFAs are one of the factors that determine the interaction between the virus and the epithelial cells of the oral cavity due to immune mechanisms and epigenetic modifications.

## 4. SCFAs and the Oral–Gut Axis

The gut’s production of SCFAs is contingent on the composition of the gut microbiota, available nutrients, and overall gut health [114]. SCFAs are absorbed within the cecum and colon through mechanisms involving proton-ion-dependent nonionic diffusion and active transport mediated by Na^+^- and H^+^-conjugated monocarboxylate transporters [115]. Within the intestine, SCFA concentrations range from 20 to 140 mM and exhibit segmental variation, with higher levels typically detected in the proximal colon [116]. The colon’s SCFA rates are higher than in the oral cavity, even in the presence of periodontitis. Nevertheless, this disparity does not result in damage to the cells of the intestinal epithelium. This apparent paradox can be elucidated by examining the structural differences between the oral and intestinal mucosa [117] (Figure 1). In contrast, the intestine is composed of a simple columnar monolayer epithelium characterized by high permeability [118]. SCFAs are actively absorbed by intestinal cells, where they are partially metabolized as an energy source and partially transported and disseminated through the bloodstream. Conversely, the oral mucosa lacks the capacity to modulate SCFA levels, thereby maintaining high local concentrations. Moreover, both the intestinal and oral mucosa are coated with a layer of mucus, a viscous secretion primarily composed of mucins. Mucus serves a protective role for epithelial tissue and can also retain specific substances. The thickness of the oral mucosa is estimated at 70–100 µm, whereas the intestinal mucosal layer comprises a loosely attached outer layer approximately 150–400 µm thick and an inner mucosal layer estimated at 800–900 µm thick [119,120]. These structural differences between the oral and intestinal mucosa indicate that the intestinal epithelium boasts a diverse array of adaptive mechanisms aimed at minimizing the adverse effects of SCFAs [121].

Thus, an appreciation of these varieties highlights the importance of context-specific considerations when studying the effects of SCFAs on different mucosal tissues.

### 4.1. Oral Bacteria and SCFAs Production in the Gut

The acidic environment of the stomach, the alkaline environment of the gut, and the abundance of digestive enzymes provide a natural barrier for oral microbes to penetrate the lower gastrointestinal tract. Another important factor is the resistance to colonization mediated by the commensal gut microbiota, which eliminates oral bacteria [122]. There are conflicting data about the possibility of colonization of the gut microbiome by oral microorganisms in healthy individuals [123]. It has been noted that pathological conditions of the stomach, such as achlorhydria, promote colonization [124]. Moreover, barrier function was found to decrease with age and the presence of diseases such as alcoholism, cirrhosis, HIV infection, inflammatory bowel disease, colorectal cancer, and rheumatoid arthritis [125,126,127,128,129,130,131]. In a murine model transplant of the human oral microbiome, it has been demonstrated that *Actinomyces, Streptococcus*, *Haemophilus*, *Veillonella*, *Fusobacterium*, and *Trichococcus* are capable of colonizing the distal colon [132].

Moreover, in a rodent model of periodontitis induced by *F. nucleatum,* researchers detected the migration of oral bacteria to the intestinal tract, concomitant with shifts in the gut microbiota. Specifically, the gut ecosystem exhibited modifications, with a notable reduction in *Verrucomicrobia* and an elevation in *Proteobacteria* evident within a two-week timeframe, followed by subsequent increases in *Bacteroidetes* and *Firmicutes* observed to the four-week time point [133]. For in vivo study on the C57BL/6 mouse strain, the amount of *Klebsiella* and *Enterobacter* species in the oral cavity and intestine increased in periodontitis; pathobionts promoted Th17 cell activation and colitis development [134]. In another in vivo study, a model of *P. gingivalis* periodontitis found an increase in *Bacteroidetes* and a decrease in *Firmicutes*, but subsequent analysis did not detect specific *P. gingivalis* DNA in the gut. In contrast, in another murine study, a single injection of *P. gingivalis* similarly altered the gut microbiota after 48 h in comparison to a control group [135,136].

Pilot clinical studies demonstrate comparable data. A clinical study of 44 participants showed that *Firmicutes*, *Proteobacteria*, *Verrucomicrobia*, and *Euryarchaeota* were increased, while *Bacteroidetes* were decreased in patients with chronic periodontitis. Representatives of *Mogibacteriaceae*, *Ruminococcaceae*, and *Prevotella* significantly differentiated individuals with oral disease from healthy controls [137]. Another study found a correlation between *Veillonella* levels in saliva and feces, indicating intestinal colonization by this microorganism; hypertension was also detected as a factor contributing to the spread of *Veillonella* in the gut [138]. *A. actinomycetemcomitans* has been shown to colonize the intestine and negatively affect lipid metabolism and insulin resistance [139]. Patients with type 1 diabetes have been shown to have decreased *S. salivarius* and increased *S. mutans*, reflected by an increased abundance of facultative anaerobes, including *Enterobacteria* [140].

Alterations in the composition of the microbiota may lead to changes in the production of SCFAs. Although there is evidence that the *Bacteroides/Firmicutes* ratio does not affect the total concentration of SCFAs, an increase in this ratio appears to lead to a relative elevation in the concentration of acetate and propionate and a decrease in butyrate [141]. Two-week oral administration of *S. mitis, S. salivarius, P. gingivalis,* or *P. nigrescens* to mice led to an increase in *Staphylococcus* and *Bacteroides* and a decrease in *Lactobacillus* spp in the gut, as well as a pronounced decrease in the level of butyric acid in the feces [142]. In human studies, *Lachnospiraceae*, a family of butyrate-producing clostridia, were associated with fewer oral bacteria in the gut, apparently improving resistance to colonization [143] in patients with colorectal cancer. *Peptostreptococcus*, *Parvimonas*, and *Fusobacterium* were associated with tumor growth in this group but were also found in healthy individuals.

From the above, it can be concluded that oral bacteria, especially opportunistic species, can colonize the intestine and affect the gut microbiome, leading to changes in the production of various metabolites, including SCFAs. This may contribute to the development of diseases of the digestive tract.

### 4.2. Oral and Systemic Effects of Gut Bacteria

Although the spread of oral microorganisms to the gut is fairly obvious due to the anatomical features of the digestive tract, the ability of gut bacteria to influence the oral microbiome is not so straightforward. Emerging evidence suggests that SCFAs produced in the gut support overall bone health. An increase in serum SCFA levels correlates directly with enhanced production of insulin-like growth factor 1 (IGF-1) and the proliferation and differentiation of regulatory T cells, which, in turn, diminish osteoclast activity and foster the production of WNT10b [144,145]. Consequently, research has shown that administration of SCFAs or implementation of a high-fiber diet can mitigate postmenopausal and inflammation-induced bone loss in murine models. Remarkably, propionate and butyrate exhibit the ability to inhibit osteoclast activity via the suppression of *NFATc1* and *TRAF6* gene expression, both in vitro and in vivo [146]. In an osteoporosis model on rats, the treatment of the gut with periodontitis patients’ saliva resulted in increased bone loss, possibly attributable to alterations in the lipolysis processes of tryptophan metabolism [147]. Considering a high-fat diet, its administration led to a reduction in *Bacteroidetes* and an increase in *Firmicutes* populations within the gut microbiota of C57BL/6J mice. This dietary shift subsequently resulted in decreased gingival blood flow and diminished blood SCFA levels, potentially culminating in long-term alveolar bone deterioration and the onset of periodontal disease [148]. Exploring the connection between intestinal microbiome composition and periodontitis risk, nine taxa were identified as associated with an increased risk of periodontitis: *Prevotella, Lachnospiraceae*, *Enterobacteriales*, *Pasteurellales*, *Enterobacteriaceae*, *Pasteurellaceae*, *Bacteroidales*, *Alistipes*, and *Eisenbergiella* groups. Conversely, *Butyricicoccus and Ruminiclostridium* were linked to a reduced risk of periodontitis [149].

Investigating the links between gut bacteria and the development of periodontitis, a clinical study demonstrated significant differences in the oral microbiome between individuals with IBD and healthy controls. The IBD group was characterized by higher levels of the *Veillonella* and *Prevotella* genera, while *Neisseria*, *Streptococcus*, *Haemophilus*, and *Fusobacterium* were associated with a healthy gut microbiome. It should be highlighted that *Streptococcus* was the sole microorganism simultaneously found in the saliva and feces of IBD patients [150].

The research data point out the complex interplay between oral and gut microbiota and their potential impact on systemic health, particularly in relation to bone health and periodontal disease. Further research projects are demanded to clarify the precise mechanisms underlying these interactions and their clinical implications.

## 5. Conclusions

The complicated relationship between the human oral microbiome, chronic inflammation, and SCFAs, as well as the health implications, is causing a growing research interest nowadays. The current research data assessing the effects of SCFAs on human health are disputable and ambiguous. When investigating the role of SCFAs in human health and disease, it is crucial to distinguish between their local GI effects and the systemic influences. Locally, SCFAs are a part of normal oral microbiota metabolism, but the increased formation of SCFAs usually attributes to dysbiosis. Moreover, excess SCFAs participate in the development of local diseases, including microbiome structure alterations, periodontal pathogen growth, a direct cytotoxic effect on the oral mucosa, consecutive damage to oral tissues, etc. On the other hand, a number of studies have established the positive impact of SCFAs on human health as a whole, including the reduction of chronic systemic inflammation and the improvement of carbohydrates and lipid metabolism. Furthermore, according to current data, serum SCFA levels are epidemiologically associated with a lower incidence of some types of cancer.

Thus, a complex and sophisticated approach with consideration of origin and localization for SCFA function assessment is demanded. Therefore, more research, especially clinical research, is needed to investigate the complicated relationship of SCFAs with health and disease and their potential role in prevention and treatment.

## Figures and Tables

**Figure 1 biomedicines-11-02749-f001:**
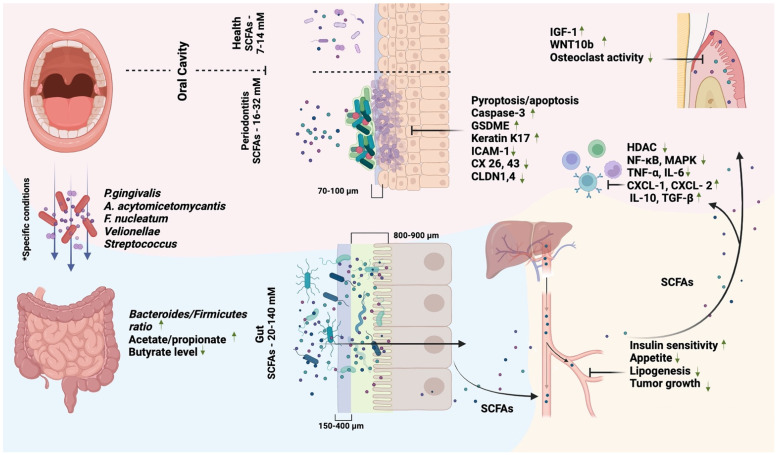
SCFAs and the oral–gut axis. SCFA-producing oral bacteria can also colonize the gut and influence the metabolism of SCFAs by the gut microbiota. The paradox of the negative effect of SCFAs in the oral cavity is demonstrated. The major contribution is made by a significantly higher thickness of the mucus layer and the type of intestinal epithelium, which provide a decrease in the local concentration of SCFAs. Increased growth of SCFA-producing bacteria and the active formation of biofilms on the surface of the oral mucosa lead to cell damage and death. SCFAs are absorbed in the intestine, enter the systemic circulation, affect glucose and lipid metabolism, reduce appetite, and have a predominantly anti-inflammatory effect. Moreover, SCFAs support osteogenesis and reduce osteoclast activity, including in the alveolar bone. * Notes. Specific conditions: digestive tract pathology (achlorhydria, cirrhosis, colorectal cancer, etc.), rheumatoid arthritis, etc. In normal physiological conditions, oral–gut microbiota colonization is still disputable.

**Table 1 biomedicines-11-02749-t001:** Ratio of SCFAs produced by certain oral bacteria [52,53,54].

Species	SCFAs				
	Acetic	Propanoic	Butyric	Isobutyric	Isovaleric
*P. gingivalis*	++	+	+++	+	++
*P. asaccharolytica*	+	+++	++	+	++
*P. intermedia*	++	+++	+	+	+
*F. nucleatum*	++	+	+++	+	+
*A. actinomycetemcomitans*	+++	-	-	-	-
*V. parvula*	++	+++	-	-	-
*S. sanguinis*	++	-	-	-	-

“+++”—relatively high concentration of SCFA in the bacterial isolate, “++”—relatively medium concentration of SCFA in the bacterial isolate, “+”—relatively low concentration of SCFA in the bacterial isolate, “-”—SCFA were not detected in the bacterial isolate.

## Data Availability

Not applicable.

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
