# Peer review of "The Complicated Relationship of Short-Chain Fatty Acids and Oral Microbiome: A Narrative Review"

_biomedicines, 2023, doi:10.3390/biomedicines11102749_

Round 1

Reviewer 1 Report

Authors presented interesting topics of role of SFA in oral  microbiome on long list of diseases.

The major weakness of presented review is simplification of complex phenomena by presence single group of compounds -  short-chain fatty acids (SCFA). It is not convincing that at lest partially transient oralbiomes SCFA might be major players on listed diseases.  It is well documented that SCFA in gut micobiome play important anti-inflammatory role  balancing pro-inflammatory components of Proteobacteria.

Another matter is lack of convincing information which described oralbiome characteristic are unique and does not belongs to all humans  microbiomes i.e.– biofilms description is typical for oralabiomes or general bacterial growth on solid surfaces ?

Long list of references are useful – be Authors  need to be more reluctant with presented studies due to by fact that majorities of them are based on few cases studies.   

In conclusion – presented manuscript need to corrected to convinced readers that SFA of oralbiomes  are solely major components of listed diseases.

Author Response

The authors very much appreciated the constructive comments on this manuscript by the reviewer. The comments have been very thorough and useful in improving the manuscript. 

- Yes, you are right. We have changed the text of the manuscript to emphasise that SCFAs are one of many possible negative factors in the development of oral disease.
- Corrected. We pointed out that biofilms are formed in other microbiomes.
- Corrected. We have highlighted the limitations of the studies reviewed.
We have substantially rewritten the manuscript and incorporated your comments as far as possible.

Reviewer 2 Report

I read the manuscript entitled “The Complicated relationship of Short Chain Fatty Acids and Oral Microbiome” with interest. However, as a review study, some minor revisions are needed in the structure of the manuscript as follows:

 - Please add the research type to the paper title.

- In the abstract section, only describes the purpose of the study, it needs to be revised to additionally describe the research method and draw conclusions based on the main research results.

- Please add a description of the type of study as a review paper.

- In the conclusion section, please add the strengths and limitations of this study.

Author Response

The authors very much appreciated the constructive comments on this manuscript by the reviewer. The comments have been very thorough and useful in improving the manuscript. 

- Corrected. We indicated the study type in the article title.
- Corrected. We have rewritten the abstract to include more details covered in this review.
- Corrected. We have highlighted the limitations of the studies reviewed.
We have substantially rewritten the manuscript and incorporated your comments as far as possible.